# Incidence of Anorexia Nervosa in Women: A Systematic Review and Meta-Analysis

**DOI:** 10.3390/ijerph17113824

**Published:** 2020-05-28

**Authors:** Leticia Martínez-González, Tania Fernández-Villa, Antonio José Molina, Miguel Delgado-Rodríguez, Vicente Martín

**Affiliations:** 1The Research Group in Gen-Environment and Health Interactions (GIIGAS), Institute of Biomedicine (IBIOMED), Universidad de León, 24071 León, Spain; leticia.martinezgonzalez6@gmail.com (L.M.-G.); ajmolt@unileon.es (A.J.M.); vmars@unileon.es (V.M.); 2Unit of Oncology, Marqués of Valdecilla University Hospital, 39008 Santander, Spain; 3Area of Preventive Medicine and Public Health, University of Jaén, 23009 Jaén, Spain; mdelgado@ujaen.es; 4The Biomedical Research Centre Network for Epidemiology and Public Health (CIBERESP), 28029 Madrid, Spain

**Keywords:** incidence, anorexia nervosa, women, epidemiology

## Abstract

*Background*: Anorexia nervosa (AN) among the general population is a rare but often fatal illness. Objective: To summarize the incidence of AN using a systematic review and meta-analysis. *Methods*: Four online databases (PubMed, Scopus, WoS and Embase) were consulted. The review was conducted according to with Preferred Reporting Items for Systematic Reviews and Meta-Analyses (PRISMA) guidelines and was limited to women. The methodological quality of the studies was assessed by the Newcastle–Ottawa Scale (NOS). *Results*: A total of 31 articles were included in the study. The incidence rate of AN ranged from 0.5 to 318.0 cases per 100,000 women–years. The incidence in studies based on outpatient healthcare services (OHS) was higher than those based on hospital admissions (HA) (8.8 95% CI: 7.83–9.80 vs. 5.0 95% CI: 4.87–5.05). In young women, the incidence in OHS was higher than HA (63.7, 95% CI 61.21–66.12 vs. 8.1 95% CI 7.60–8.53). The linear trend in the incidence of AN was increasing in all ages of women and young women, both in studies with hospital admission records, and in those based on outpatient healthcare services. *Conclusion*: The incidence of AN depends on the methodology, the type of population and the diagnostic criteria used.

## 1. Introduction

Anorexia Nervosa (AN) is a psychiatric disorder characterized by both physical and psychological symptoms. This disease occurs more frequently in young women in Western countries, although it is also diagnosed in older women in non-Western countries [1]. There is extensive variability in the incidence of AN in the published reports, and there is also a controversy about the trend in the incidence of AN in recent decades. A meta-analysis on the incidence of AN in mental health care reported an increase in Europe until the 1970s [2], and a stabilization thereafter [3,4]. However, incidence rates of the first hospitalization for AN in women are increasing in recently published studies [5,6].

This variability could be due, at least in part, to the improvements in case detection over time [1]. In addition, we must also bear in mind the fact that incidence can be obtained through different methodological approaches, such as cohort studies or studies based on medical records from outpatient healthcare services (OHS) or patient hospital admissions (HA). Furthermore, different case identification systems and diagnostic criteria, based on the different updates of International Classification of Diseases (ICD) or the Diagnostic and Statistical Manual of Mental Disorder (DSM) [7,8,9,10,11,12,13,14,15], have been used, which may have generated more heterogeneity among the studies.

This heterogeneity makes it difficult to know the actual incidence of AN and its trend over time. The aim of this review was thus to systematically review and perform a meta-analysis of the literature on the incidence of AN in women published between the period 1980 and 2019, especially in young women, taking into account the different methodological approaches that can be used.

## 2. Materials and Methods

### 2.1. Search Strategy

The review was conducted in accordance with Preferred Reporting Items for Systematic Reviews and Meta-Analyses (PRISMA) [16] guidelines. The literature was reviewed from 1980 to December 2019 using the following databases: PubMed, Scopus, Web of Science, and Embase. Eligibility was established by two researchers reading the title, abstract and full text of each study. A detailed search strategy can be seen in Appendix A. 

### 2.2. Eligibility Criteria

We collected original articles without a language restriction that met the following inclusion criteria:Publications presenting AN incidence in women based on Outpatient Healthcare Services (OHS), Hospital Admissions (HA) or cohorts that referred to the general population (not subgroups such as pregnant women or women working in the military);Reports based on hospital admissions that also required AN to be the primary diagnosis; only the first admission was considered;Studies detailing the methods used to assess incidence;Case ascertainments applying “narrow” AN, which means a strict diagnostic criterion. “Narrow AN” includes only full-criteria AN (eg., Feighner, ICD-9 307B, ICD-10 F50.0, and DSM-IV AN cases [17].

### 2.3. Exclusion Criteria

We excluded reviews, editorials, books, book chapters and commentaries. Articles were also discarded if their data were a combination of other eating disorders (e.g., AN + bulimia nervosa), or pooled both sexes without giving specific data for women. The “broad AN”, which includes both full-criteria AN and atypical AN (e.g., ICD–10 F50.1) cases, were excluded [17,18]. 

### 2.4. Title, Abstract and Article Selection

Firstly, a selection of potentially eligible articles was carried out independently by two authors (LMG, TFV) selecting articles eligible by title in the first step, then articles eligible by abstract, and finally, the literature eligible by full text. Following PRISMA guidelines, full text articles were evaluated for eligibility by two authors (LMG, TFV) and discrepancies were assessed by a third researcher (VMS). If there were several reports on a single study, only the one with the longest follow-up was included [19,20,21]. The different steps of this selection procedure are shown in the flow-chart (Figure 1).

### 2.5. Quality Assessment of Primary Studies

The Newcastle–Ottawa Scale for the Assessment of Quality (NOS) establishes nine items, giving a point to each accomplished item, to classify the studies as high quality (score 7–9), moderate quality (score 4–6), or poor quality (score 0–3) [22]. This scale was applied by two independent researchers (LMG and TFV) and the discrepancies among the reviewers were resolved by a third researcher who decided on the final score shown in Appendix A.

### 2.6. Data Extraction

Information was gathered on the following variables: data collection period, location, women’s age, diagnostic criteria, number of cases and in the population, incidence rate, and 95% confidence intervals (CI). The study design was classified in three types according to the case identification: -Hospital admissions (HA): cases were identified by first admission into a psychiatric unit/hospital or general hospitals;-Outpatient healthcare services (OHS): cases were identified through medical consultations not requiring hospital admission, whether in primary care or in outpatient psychiatric units;-Cohort studies: diagnosed cases in cohort studies.

The authors of the included studies were contacted if any of the previous data were lacking.

### 2.7. Sensitivity and Grouping Criteria

In order to reduce heterogeneity and obtain more accurate results when performing meta-analyses, some sensitivity and grouping criteria for different parameters were taken into account in this work, such as the quality of the articles, the type of record, the age ranges, and the ethnic group of the population. Firstly, the low-quality studies were considered in the review but excluded from the meta-analysis. Then, we grouped the articles in HA and OHS according to the definitions explained in the previous paragraph. Then, two types of studies were selected according to age range, those referring young women (10–30 years) and those studies including all ages, while studies that reported other ranges (e.g., 5–64, 10–49) were not considered for meta-analysis. As regards the origin of the population, most of the articles include European or USA populations, so the scarce papers reporting data from other populations (Latin American or Asian) were not included in meta-analysis to reduce heterogeneity.

### 2.8. Statistical Analysis

#### 2.8.1. Meta-Analysis

Incidence rates of the primary studies, selected according to the grouping and sensitivity criteria, were pooled using the inverse-variance method. The fixed effects model was applied because it assumes that heterogeneity exists and does not want to be suppressed, especially when there are studies with large differences in sample size and population from very different cultural environments. In the studies not showing 95% CI for incidence rates, the limits were estimated using the Poisson distribution [23]. To assess heterogeneity, the Q (*p* < 0.1 for statistical significance) and I^2^ (proportion of total variability between studies) statistics were applied. Statistical analyses were performed using the “metan” command of STATA 13.0 [24]. 

#### 2.8.2. Temporal Trends Analysis

Finally, we calculated the linear trends in women of all ages and young women based on HA and OHS records, from 1940 until now. When incidences were reported for a range of years, the center of the interval was considered as the reference year for the trend calculation. 

## 3. Results

A total of 918 articles were found in the databases consulted, and 31 articles that satisfied the inclusion criteria were finally included in the study (Figure 1). 

### 3.1. General Incidence of AN

The incidence of AN in our results ranges from 0.5 to 318.9 cases per 100,000 women–years, and in young women from 0.6 to 37.1 cases per 100,000 women–years, with significant variations depending on the source of the cases.

### 3.2. Incidence of AN in Hospital Admissions

A total number of 12 articles based on hospital records were included, with the incidence of AN ranging between 0.5 and 7.5 cases per 100,000 women–years (Appendix A). Five studies included all ages of women [6,25,26,27,28], but only three were chosen on the basis of the sensitivity criteria as shown in Figure 2, with a pooled incidence of 5.0 cases per 100,000 women-years; 95% CI: 4.87–5.05) but with a high heterogeneity (I^2^ = 97.4%; *p* < 0.001) [6,25,26]. Insofar as hospital-based studies with young women are concerned, ten articles [2,27,28,29,30,31,32,33,34,35] included information on the incidence in women between 10 and 29 years of age with an incidence per 100,000 women–years ranging from 0.6 to 37.1 (Appendix A). The meta-analysis of the eight studies that meet the above criteria obtained a pooled incidence of 8.1 cases per 100,000 women-years (95% CI: 7.60–8.53) with significant heterogeneity (I^2^ = 91.9%; *p* < 0.001) (Figure 3). 

### 3.3. Incidence of AN in Outpatient Healthcare Services

A total of seventeen studies based on OHS were analyzed, of which 14 provided information from women of all ages and 12 from women aged 10–29. The studies about women of all ages, provided information on the incidence of AN with a range between 2.1 and 42.3 cases per 100,000 women–years (Appendix A). A pooled incidence of 8.8 per 100,000 women–years was obtained (95% CI: 7.83–9.80) with a high heterogeneity (I^2^ = 83.5%; *p* < 0.001) (Figure 4) from the only five studies that meet the sensitivity criteria [36,37,38,39,40], since the other studies should be excluded due to their low quality [41], employment of different age ranges [41,42], or for being from geographical areas not included [42,43,44].

In young women, twelve studies provided information on the incidence of AN with a range between 3.3 [42] and 101.0 [45] per 100,000 women–years (Appendix A). Figure 5 shows the pooled analysis of the nine studies [5,36,37,39,46,47,48,49,50] included according to sensitivity criteria with an incidence of 63.7 cases per 100,000 women–years (95% CI: 61.21–66.12; I^2^ = 96.0%; *p* < 0.001). 

### 3.4. Incidence of AN in Cohort Studies

Three cohort studies were found in which the incidence of AN ranged between 120.0 and 318.9 cases per 100,000 women–years (Appendix A) and showed some differences in the sample profiles and the screening and diagnosis procedures. Thus, the study by Keski-Rakhonen et al. 2007 [51] administered a screening questionnaire to twin women and made a diagnosis using the structured clinical interview for DSM-IV (SCID) [52], while in the study by Ghaderi and Scott, 1998 [53], a screening questionnaire was also administered to the sample, though they did not specify whether the diagnosis was carried out by a doctor. Lastly, in the study from Lahortiga-Ramos et al. 2005 [54], a screening questionnaire was administered to young women and the diagnosis was done by a psychiatrist. 

### 3.5. Incidence of AN as Defined by the Measurement Tool: ICD or DSM

In Appendix A, we can also see how most of the articles analyzed use the ICD and DSM classifications, showing great variability in the incidence rates of AN depending on the classification used: DSM-III (4.0–75.5), DSM-IV (3.4–66.1), ICD-8 (1.7–12.0), ICD-9 (2.0–7.1) and ICD-10 (4.2–101.0). 

In HA studies, two studies used the DSM-III, one of them used the revised version (DSM-III-TR) where the incidences of both vary between 4.0 and 19.7 cases per 100,000 women–years. Six studies used the ICD-8 in which the incidences ranged from 1.7 to 12.0 cases per 100,000 women-years. Three studies used the ICD-9, in which the incidences ranged between 2.2 and 37.1 cases per 100,000 women–years. One study used ICD-9 and ICD-10 where the incidence was 4.3 cases per 100,000 women–years. Lastly, two studies used the ICD-10, and the incidence varied between 4.2 and 7.5 cases per 100,000 women-years.

However, in studies with a diagnosis in OHS facilities, three used the DSM-III, two of which used the revised version, and the incidence varied between 4.9 and 75.5 cases per 100,000 women-years. Four articles used the DSM-IV, whose incidence varied between 3.4 and 66.1 cases per 100,000 women-years. Two studies used the revised version (ICD-9-CM) where the incidence was 2.0–4.1 cases per 100,000 women-years. Finally, the ICD-10 was used by five articles and the incidence ranged from 7.8 to 101.0 cases per 100,000 women-years.

### 3.6. Temporal Trends in Incidences

Lastly, the analysis of the temporal trends in incidences separately for HA and OHS records are shown for women of all ages (Figure 6) and young women (Figure 7). Increasing trends in the incidence of AN were observed in both, young and all-ages women, independently of the type of records, but it is noteworthy that the upward trend is much more pronounced in studies based on OHS records in both age groups. All trends were statistically significant (*p* < 0.05), except the trend on HA for young women (*p* = 0.154).

## 4. Discussion

This article has reviewed studies published from the period 1980 to 2019 that have evaluated the incidence of AN in women. A great variability has been observed in the incidences reported according to the type of study and the source of the cases. The studies reporting the highest incidence were cohort studies, followed by studies based on OHS and finally studies based on HA.

Estimating the true incidence of AN in a population is a difficult task, due to the stigma associated with mental illness and admission to these units, among other reasons, which is why what is expected is an underestimation [2,4,50,55,56].

In cohort studies, much higher incidence rates are observed, ranging from 120.0 to 318.9 cases per 100,000 women–years [51,53,54]. These higher values are to be expected considering that, on the one hand, samples of adolescent and young women were used, and, on the other hand, screening strategies were carried out to detect the cases.

### 4.1. Incidence of AN in Hospital Admission and Outpatient Healthcare Services

The differences observed between the studies based on HA and those based on OHS are consistent with the forecast that there would be fewer, but more severe, cases in HA than in OHS [55]. Some authors estimate that less than half of AN cases are admitted to hospital units while the rest are treated in non-hospital units [39,55], which is consistent with the differences observed in our study in terms of global incidence, though this is more marked in the case of young women. This may be justified, because primary care is the first place where patients seek care before being referred to a specialized center with a psychiatrist or HA.

### 4.2. Incidence of AN in Young Women

Another finding of this study is higher incidence of AN in young women, a fact already known and reported by numerous authors [1,3,4,57,58,59,60]. The explanations given for this result are varied, mainly related to special vulnerability to the barely achievable models and beauty standards conveyed by the media, as well as the physical, hormonal and emotional changes that take place in adolescence. This search for the perfect body causes an imbalance that leads to restrictive diets and eating behaviors [61,62]. The age of greatest risk for developing AN is between 10 and 24 years in women [55,63,64,65].

It is striking that the difference between the incidence in all women and in young women is much more marked in studies based on OHS [8.8 (95% IC: 7.83–9.80) vs. 63.7 (95% IC: 61.21–66.12)] than those based on HA [5.0 (95% IC: 4.87–5.05) vs. 8.1 (95% IC: 7.60–8.53)]. This could be justified by the increased awareness of the disease on the part of health personnel and patients, given that an outpatient setting is the first treatment option in most cases. Young women go to medical centers in the early stages of the disease to be treated in these centers and hospitalization is often unnecessary. In turn, better identification of cases and perhaps greater public awareness of the problem and decreased stigma could prevent hospitalization in many cases [66].

### 4.3. Trends in the Incidence of AN

We have observed increasing temporal trends in the incidence of anorexia regardless of the type of record used or the age range analysed, whereas previous studies have recorded an increase in AN up to 1970 [55] and a stabilization in the 1990s [33,40]. Our results should be assessed with caution given the large number of methodological limitations, possible biases and sources of variability, especially the large variability in diagnostic systems [34], demographic characteristics [21,67], and readmission rates in care services [30,67].

It is interesting to note that the trend is more pronounced in records from OHS than in hospital records. This is probably due to the fact that greater knowledge of the disease and its high mortality rate compared to other mental diseases [68,69,70,71] has generated a greater capacity for detection, changes in clinical practice with the appearance of specialized services [72], and the greater social relevance of the problem [3,4,73], that has led to a greater number of diagnoses in less advanced stages, while the number of cases requiring hospitalization is less influenced by these factors and the detection of cases in early stages limits the evolution of these cases by reducing the severity of the cases and their need for hospitalization [74]. On the other hand, while for the OHS records, the slope in the evolution of the incidence among young women is much more marked than for the total of women, when we analyze the data based on hospital cases, the trends are much more similar. Among the causes that could explain this, there is a greater effort in the search for cases among adolescents [3,4,73] and a worse prognosis than in adult cases [75,76,77].

### 4.4. AN Measurement Tools

Another relevant methodological aspect has been the use of different diagnostic criteria (Feighner, ICD, DSM), which have undergone important modifications over time, as detailed below. One of the criteria that has been adjusted refers to the percentage of weight that was lower than the normal percentage anticipated. The percentage of weight lower than normal corresponds to 25% [8,78]. In people under 18 years of age, the initial weight loss should be added to the corresponding weight to be gained, in line with the growth process and should be checked to see if the sum of the two reaches 25%. In the DSM-III-R, weight loss up to 15% below the theoretical weight or failure to achieve weight gain while waiting out the growth phase. In the DSM-IV and DSM-IV-TR, underweight is established to be a weight less than 85% of the normal weight, taking into account age and height. Another important diagnostic criterion is the amenorrhea criterion, defined as the absence of at least three consecutive menstrual cycles, which has been preserved from DMS-III-R to DSM-IV.

There are few studies that compare the consistency of the different AN classifications [79,80,81,82]. In our work on young women, the DSM-IV diagnosed a greater number of cases than the ICD-10, coinciding with another hospital study conducted on children [79]. Our results show that, in young women, the ICD-10 diagnosed fewer cases than the DSM-III-TR, which does not coincide with a study conducted in 18-year-old women where the ICD-10 diagnosed more cases than the DSM-III-TR [80]. As such, we must bear in mind that, in our work, it is young women who go to medical centers vs. school-aged girls [83]. This could be justified because the DSM-IV and DSM-III-TR present a less stringent diagnostic criteria regarding weight, since both determine a weight less than 85% of the normal, considering age and height, while ICD-10 weight loss corresponds to at least 15% below the normal weight expected for the age and the corresponding size. Another study finds that there is sound agreement between ICD-9 and DSM-III in hospitalized adolescents [81]. In our work, the incidence in young women hospitalized in the ICD-9 is higher than the DSM-III-TR. Since there are large differences between the DSM-III and the DSM-III-TR, it is not possible to compare them with the aforementioned study. The last article in outpatient children and adolescents diagnosed more cases with ICD-9 than with ICD-10 [82].

Some studies, like the one carried out by Milos et al. 2006 [33], used the diagnostic criteria of Feighner et al. 1972 [78]. These criteria are very restrictive and could have excluded a relevant number of patients with AN. The criteria have low incidences of AN and, in addition, have been used in a study of HA, which has led to further underestimation. Another work calculates incidence from the clinical diagnoses of primary care physicians rather than DSM-IV [40].

### 4.5. AN in Other Cultures

An important aspect to take into account is the scarce number of epidemiological studies that were carried out in non-Western countries [84,85], an extrapolation problem in which only studies of AN incidence in populations from Europe, the USA and Australia [86] were done. However, the prevalence of AN is increasing in Asian, Arab and Pacific regions caused by increasing industrialization, urbanization, and globalization [87,88]. Some studies show that there are cases of AN in Latin America, but no cases of AN have been found in black women. A systematic review with meta-analysis [57] shows that the prevalence of AN is much higher in European countries than in South America, explaining that this could be due to differences in body type between cultures [89,90,91] and could be a protection factor in young people [59]. Lastly, this review showed that AN most frequently occurs in the high-risk group of young Western females, but can also occur in older women and in men [1].

### 4.6. Strengths and Limitations of the Study

The main advantage of this study is the exhaustive bibliographic search of the existing literature from 1980 to 31 December 2019, followed by a systematic review and meta-analysis. A large number of articles were studied, which means that arguably the most relevant studies evaluating the incidence of AN were included in this review. We must highlight the difficulty in carrying out this review due to the heterogeneity of the studies, even in simple matters like using standard age categories for all studies, which is the exception rather than the norm. Another main difficulty was the wide range of instruments used to assess the incidence rates of AN. The included studies vary greatly depending on the instruments used, the sample and the study design. Therefore, given the heterogeneity, the incidence rates of AN reported in the meta-analyses remain difficult to interpret.

Furthermore, other limitations involve performing studies on AN in women in Europe and the United States and three studies in other populations (Curaçao and Taiwan), as well as focusing on young people between 15 and 25 years old, excluding other age groups. While it would be interesting for future research to evaluate the incidence of AN in other populations (Eastern cultures, Latin America, etc.), it would also be interesting to know if the incidence changes in other age cohorts.

## 5. Conclusions

In conclusion, the current literature is characterized by significant variations in the methods used to assess the incidence of AN and the wide variability of the diagnostic criteria. Consequently, incidence rates vary substantially among published studies and, because they are highly influenced by the tools used to measure them, methodological data and research results should be taken into account when interpreting AN incidence rates. The general trends in the incidence of AN in women are increasing, especially in young women, though this may be largely due to changes in the way medical services approach this pathology and the sensitivity of the population.

The results of this study provide an overview of the evolution and current situation of AN in women, which is especially relevant for early diagnosis of women and the support of early recovery and even prevention of the disease, through awareness raising among the population and health professionals. Future epidemiological research should move towards the identification of specific risk factors for the AN, and towards the unification of diagnostic criteria for better approximation of trends.

## Figures and Tables

**Figure 1 ijerph-17-03824-f001:**
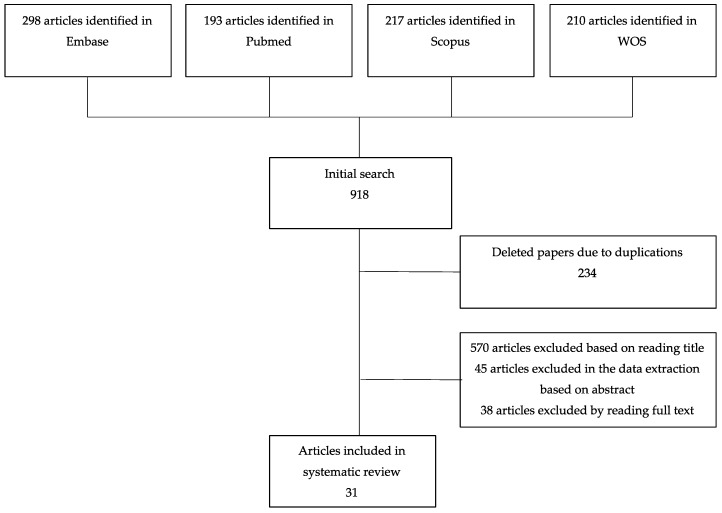
Flow-chart of selection process.

**Figure 2 ijerph-17-03824-f002:**
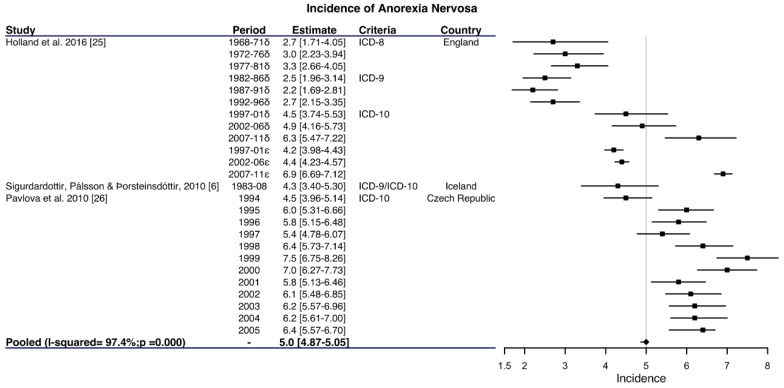
Forest plot of the meta-analysis of the anorexia nervosa incidence rates in women in hospital admissions. ^δ^ Average annual age-standardized hospital first recorded admission rates for anorexia nervosa per 100,000 women-years in Oxford and West Berkshire (England). ^ε^ Average annual age-standardized hospital first recorded admission rates for anorexia nervosa per 100,000 women–years in England.

**Figure 3 ijerph-17-03824-f003:**
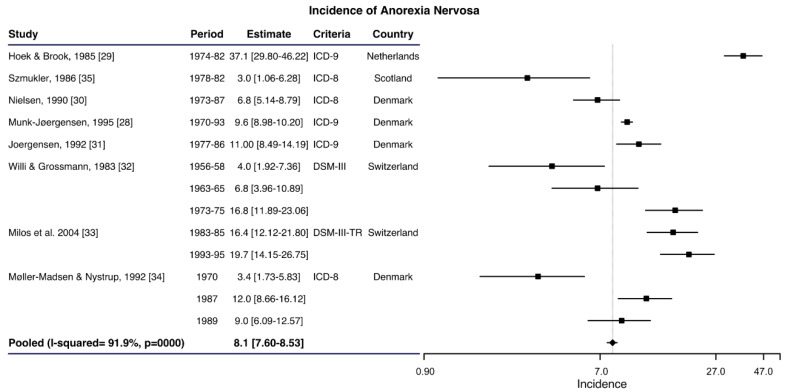
Forest plot of the meta-analysis of the anorexia nervosa incidence rates in young women in hospital admissions.

**Figure 4 ijerph-17-03824-f004:**
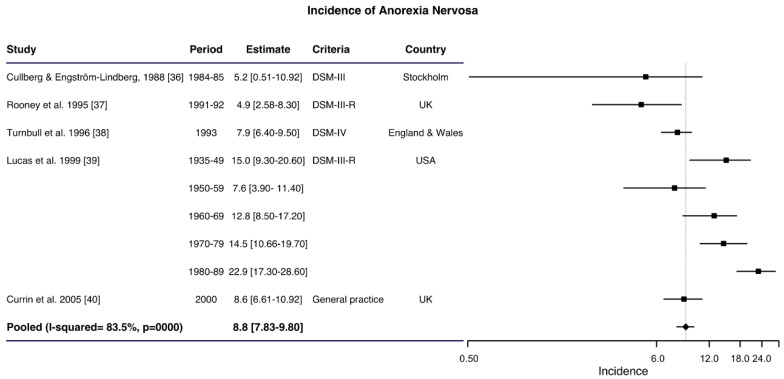
Forest plot of the meta-analysis of anorexia nervosa incidence rates in women in outpatient healthcare services.

**Figure 5 ijerph-17-03824-f005:**
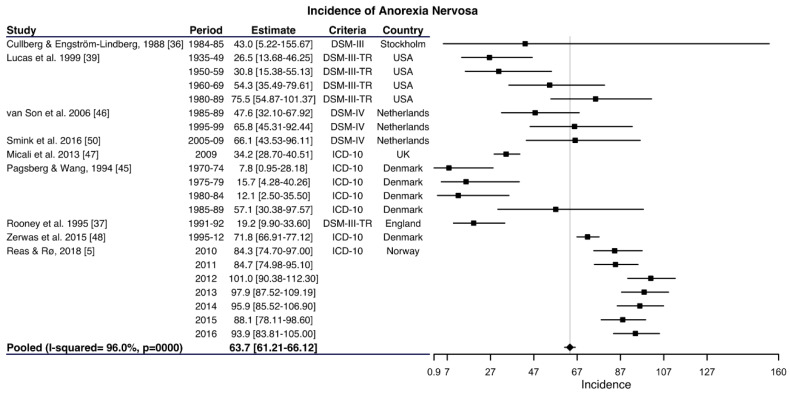
Forest plot of the meta-analysis of anorexia nervosa incidence rates in young women in outpatient healthcare services.

**Figure 6 ijerph-17-03824-f006:**
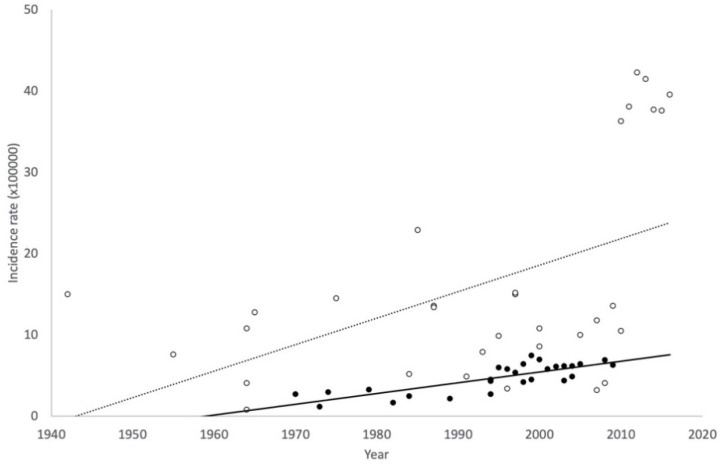
Trends in the incidence of anorexia nervosa in women of all ages for hospital admission (HA) studies (Black circle and continuous line) and outpatient healthcare services (OHS) studies (White circle and dotted line).

**Figure 7 ijerph-17-03824-f007:**
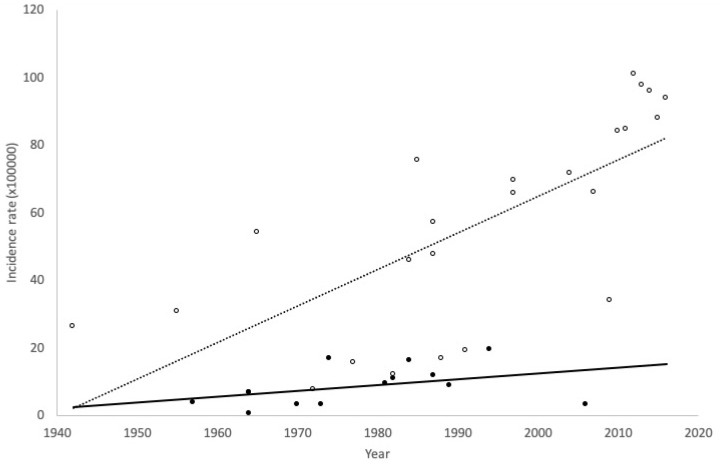
Trends in the incidence of anorexia nervosa in young women for hospital admission (HA) studies (Black circle and continuous line) and outpatient healthcare services (OHS) studies (White circle and dotted line).

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
