# Peer review of "Incidence of Anorexia Nervosa in Women: A Systematic Review and Meta-Analysis"

_ijerph, 2020, doi:10.3390/ijerph17113824_

Round 1

Reviewer 1 Report

The manuscript “Incidence of anorexia nervosa in women: a systematic review and meta-analysis” by Martínez-González et al. is a well-described and well-written article.

I have the following minor comments:

  • Line 52: The weblink is taking to the University webpage. The guideline could not be found there. Please check the hyperlink. Maybe the reference can be put in the reference list at the end and not as the weblink.
  • Line 53: “The literature was reviewed up to December 2019”, but the start date has not been mentioned here. (The start date as 1980 has been mentioned in the limitation section 4.6. The authors might consider mentioning it in the method section).
  • Figure 6: The p values for the trends are not discussed. Were those significant (p<0.05)?

Author Response

Dear reviewer,

Thank you very much for your contributions. We have made the following changes in response to your comments:

  • Line 52: The weblink is taking to the University webpage. The guideline could not be found there. Please check the hyperlink. Maybe the reference can be put in the reference list at the end and not as the weblink.

Thanks for the comment.We have removed the web page from the text, because it does not actually lead to the PRISMA guide citation, which is referenced in the biliography section as number 16.

  • Line 53: “The literature was reviewed up to December 2019”, but the start date has not been mentioned here. (The start date as 1980 has been mentioned in the limitation section 4.6. The authors might consider mentioning it in the method section).

Thank you very much for this comment, we have made a mistake not mentioning the beginning date of the search in the methodology section. We have added the following: The literature was reviewed from 1980 to December 2019…

  • Figure 6: The p values for the trends are not discussed. Were those significant (p<0.05)?

Thank you again for this comment. Because of a mistake, we forgot to include the statistical significance. All trends were significant (p<0.05) except the HA trend in young people whose value of p=0.154. These appreciations have been incorporated into the document.

Reviewer 2 Report

In the present study, the authors report the incidence of anorexia nervosa in a women cohort. The study is very interesting, a large number of articles have been investigated and the results are well presented. I have one minor comment that has to be addressed by the authors.

Can these findings help the community face the problem of the incidence of anorexia nervosa in women or even prevent it? Please refer extensively to the physiological meaning of the results of this study. 

Author Response

Dear reviewer,

Thank you very much for your comment. We have increased the final conclusion incorporating the commentary that you ask us with the following text:

“The results of this study provide an overview of the evolution and current situation of AN in women, which is especially relevant for early diagnosis of women and the support of early recovery and even prevention of the disease, through awareness raising among the population and health professionals. Future epidemiological research should move towards the identification of specific risk factors for the AN, towards the unification of diagnostic criteria for better approximation of trends”.

Reviewer 3 Report

The paper covers a very important problem in psychiatry. The authors faced the challenge to compare incidence of anorexia nervosa in women during a long period of time. The methods are well chosen and described. The discussion and conclusions are short, but they correspond with the results. The paper has its limitations, e.g. different diagnosis criteria that were changing during a long period of time covered by the analysis, however it was pointed in the limitations of the study. 

My only concern refers to the literature - it does not follow all of the journal formatting rules.

The is also pages numbers information missing for literature positions no. 7-14. 

Author Response

Dear reviewer,

Thank you very much for your appreciation. We have reviewed all the bibliographic citations. Thanks to your commentary we have observed 3 duplicate citations. Therefore, we have not only changed the format, but also eliminated the duplicate citations and modified the numbers of the references throughout the document, in the tables, the figures and the final list.

We have not included all the page numbers of the citations you mention, because they are books that we have consulted in our university library, which is closed due to confinement because of COVID-19. Therefore, we have not been able to access them. Sorry.